# Sexualized Images on Social Media and Adolescent Girls’ Mental Health: Qualitative Insights from Parents, School Support Service Staff and Youth Mental Health Service Providers

**DOI:** 10.3390/ijerph20010433

**Published:** 2022-12-27

**Authors:** Alana Papageorgiou, Donna Cross, Colleen Fisher

**Affiliations:** 1Telethon Kids Institute, The University of Western Australia, Perth 6009, Australia; 2School of Population and Global Health, The University of Western Australia, Perth 6009, Australia

**Keywords:** adolescence, body image, self-esteem, gender, comparisons, support, social media use

## Abstract

This research explored adults’ perceptions of how sexualized images typically found on social media might influence adolescent girls’ mental health, what support girls might need should they experience mental health difficulties, and how such difficulties could be prevented or reduced. Qualitative data were collected using semi-structured in-depth interviews with parents of adolescent girls (n = 11) and those who provide support to them: school support service staff (n = 7) and youth mental health service providers (n = 10) located in Perth, Western Australia. All three participant groups perceived sexualized images typically found on social media as exacerbating poor mental health among adolescent girls. Two interrelated themes, emerged with participants describing the ‘potential for comparison’ and ‘pressure to conform’ they believed girls encounter on social media that influences their mental health. Participants also explained how they perceived ‘counteracting negative influences’ related to sexualized images on social media could prevent or reduce the potential for mental health harms among girls, and the importance of adults and services ‘keeping up to date’ and being ‘approachable and trustworthy’ when describing the support they believed girls might need. The findings of this study have important implications for the development of health promotion programs focused on social media use and mental health among adolescent girls.

## 1. Introduction

Adolescence is a developmental period critical to identity formation, with considerable physical, social, cognitive, and emotional changes [1]. For adolescents in today’s digital age, the online environment including social media plays a central role in how they socialize and connect with peers and form their identity and sense of self [2]. Social media platforms such as Instagram, Snapchat and TikTok focus on users creating and sharing photos and videos that enable instant social interaction [3,4]. Adolescents are avid social media users, with 63% of young people aged 13 to 18 years in the United States (US) using social media daily [4] and 85% having at least one social media account by the age of 14 years [5]. Adolescent girls use social media frequently, with 70% of 13–18 year old girls in the US reporting they use social media everyday compared to 56% of boys [4]. Comparably, adolescent girls and young women aged 14 to 24 years in Australia spend nearly two hours per day on average using social media and almost five hours more per week compared to same-aged males [6]. The impacts of such widespread and frequent social media use among adolescents have been found to contribute to their lives in both positive and negative ways. The benefits of social media use include enhanced opportunities for learning and greater peer connection and support [7,8,9]. More widely documented, however, are the adverse influences from adolescents’ social media use.

Several quantitative studies suggest that social media use is associated with adolescents’ mental health difficulties including increased depression, anxiety, and self-harm behaviors, and lower socio-emotional wellbeing, low self-esteem and negative body image [10,11,12,13]. Qualitative findings from focus groups with adolescent girls exploring social media use and body image suggest that they use social media to engage in appearance-focused social comparisons [14,15], potentially magnifying this negative impact for adolescent girls compared to adolescent boys.

The sexualization of females using social media is a factor that may negatively influence adolescent girls’ mental health. Sexualization occurs when a person is: valued for their sex appeal over other characteristics; held to a standard where narrowly defined physical attractiveness is equated with being sexy; sexually objectified (considered an object for others’ sexual use rather than having capacity for independent action and decision making); and/or has sexuality inappropriately imposed on them [16]. Studies exploring the impact of sexualized images of females in traditional forms of media including television, film and magazines suggest such images are associated with depressive symptoms and negative body image among adolescent girls [17,18]. Social media, however, is now the more prominent form of media used by adolescent girls and features an abundance of images portraying girls and women as sexually available and objectified that emphasize sexual attractiveness and physical appearance [18,19,20]. Additionally, unlike traditional media, adolescent girls can create and share sexualized images of themselves on social media and gain instant feedback on these images from their peers, whose sexualized images they can also view [21,22]. This has resulted in increased research investigating how exposure to sexualized images in social media, might influence girls’ mental health.

When adolescent girls view sexualized images while using social media, they may internalize an observer’s perspective as a primary view of themselves and their body as an object valued for its appearance, known as ‘self-objectification’ [17,20,23]. A recent systematic review suggests that self-objectification is associated with depressive symptoms and disordered eating among girls aged 18 years and younger [24]. A meta-analysis also found sexualizing online media use (measured as using the Internet or social media) increased self-objectification among adolescent girls [25]. Furthermore, the frequency of viewing sexualized images on Instagram was found longitudinally to predict self-objectification among adolescents, with girls reporting subsequent body image concerns [26]. Hence, there appears to be a relationship between sexualized images on social media and self-objectification among girls. This association may be explained by Social Cognitive Theory [27], where the endorsement of such images through ‘likes’ and comments teaches girls what are acceptable social norms related to sexualized behavior online. Sexualized images of females are rewarded on social media and have been found to receive more positive feedback compared to less sexualized images, reinforcing the value of girls’ physical appearance [21,28]. The influence of sexualized images on social media on adolescent girls’ mental health may therefore involve an interplay between self-objectification and social norms.

Qualitative research with adolescent girls has found they frequently encounter appearance-focused and objectified images of females when using social media [29,30,31]. Additionally, qualitative research has highlighted how these sexualized ideals of femininity portrayed in such images appear to influence girls’ self-presentations online [29,32,33]. In a qualitative study exploring adolescent girls’ perceptions of sexualized images of females on social media, girls reported that conforming to gendered and social norms creates an expectation that they post sexualized images of themselves on social media for peer acceptance and approval [31]. The resulting appearance-based comparisons made with peers, however, appear to influence poor body image [14,15,34]. The potential negative impacts of sexualized images on social media for adolescent girls’ mental health highlights the need for support and prevention and early intervention to help minimize any harms to their wellbeing. As adolescent girls navigate the social media environment and other socio-cultural constructs that portray females as sexual, during this critical time in their development, it is crucial for the adults in their lives to provide appropriate support.

Families, school support service staff and youth mental health service providers are important sources of support for adolescents’ mental health and their inclusion in effective prevention and intervention strategies is highly warranted [21,30,35,36,37]. Qualitative research exploring secondary school teachers’ and school support service staff members’ experiences of the role of social media in relation to the mental health of their students found they perceived social media as having a negative effect on students’ mental health, through social comparison and body pressure, reinforcing negative emotions associated with acceptance and rejection, motivating self-presentation [38]. However, no published research was found describing parental or school or community-based youth mental health service providers’ views about how sexualized images on social media might influence adolescent girls’ mental health. Furthermore, no evidence was found describing what support adolescent girls may need if they experience mental health difficulties from their exposure to sexualized images on social media, or how such difficulties could be prevented or minimized. Such perspectives are important to guide the development of appropriate health promotion programs. Therefore, the aim of this study was to explore the perceptions of parents, school support service staff, and youth mental health service providers of how sexualized images on social media might influence girls’ mental health (in positive or negative ways), what support girls might need should they experience mental health difficulties, and how such difficulties could be prevented or reduced. This paper forms part of a study that explored the same research questions but from the perspective of adolescent girls.

## 2. Materials and Methods

### 2.1. Research Design

A generic qualitative approach [39], underpinned by a constructivist epistemology which recognises that individuals give meaning to their own unique realities [40,41], using one-on-one in-depth interviews was utilised for this study.

### 2.2. Participants

The study comprised a purposive sample of three participant groups based in Perth, Western Australia; parents, school support service staff, and youth mental health service providers—to ensure a wide range of perspectives from adults who support adolescent girls. Eleven parents of daughters aged 14–17 years (Grades 9–11) were recruited using central intercept at sporting clubs (n = 8) and snowball sampling techniques (n = 3). Parents of adolescent girls participating in the broader study were excluded to avoid potential family conflict resulting from family groups being in a study addressing sensitive topics. Seven school support service staff including school psychologists and those in a student support role were recruited from metropolitan government (n = 2) and co-educational (n = 1) and all girls’ (n = 4) non-government schools. Ten youth mental health service providers were recruited from external agencies providing support to girls in clinical settings (n = 8) and through snowball sampling techniques (n = 2).

### 2.3. Recruitment

Three participant groups were recruited for this study: parents, school support service staff, and youth mental health service providers. To recruit parents from sporting clubs, approval was sought from sporting associations, allowing the first author to attend sporting club centers on game days. Parents attending the games were provided with study information and asked if they had a daughter/s between 14 and 17 years of age. Both female and male parents were invited to participate. Those interested in participating completed an expression of interest, sharing their contact details and were provided with information about the study. Those who completed an expression of interest were subsequently contacted to arrange a date and time for an interview. None of the male parents approached during recruitment agreed to participate in the study. To recruit school support service staff, approval was obtained from the relevant government and non-government school sectors. The principal from each school was contacted, seeking their approval for project information to be emailed to staff in a support service role within their school. Youth mental health services were emailed study information and asked to distribute to individual service providers within their service. Once a youth mental health service provider indicated his/her interest in participating in the study, a participant information form with further information was emailed and a date and time for the interview arranged by the first author.

Snowball sampling methods were also used to recruit all participant groups. Study participants were asked to distribute project information to parents with a daughter/s aged between 14 and 17 years of age; other school support service staff; other youth mental health service providers, and within their professional networks.

### 2.4. Data Collection

One-on-one semi-structured interviews, using an interview guide with open-ended questioning were undertaken with each participant group by the first author. To understand individual perspectives and experiences from participants in each group, questions were exploratory, allowing participants to provide descriptions of their experiences, thoughts and views using a recursive, conversational approach to maintain participant comfort during the interview [42]. Prior to interview commencement, study information was discussed, providing participants the opportunity to ask questions to seek clarification or further information. Participants signed a consent form, agreeing to their participation and the interview being audio recorded. Prior to data collection, the interview guide was pilot tested with a convenience sample of two parents/carers of adolescent girls, one child health psychologist and two school support service staff. No changes were required from the pilot testing.

During the interviews, participants were asked about their observations and perceptions of how the sexualization of females through images on social media might influence adolescent girls’ mental health and in what ways these could be positive or negative. If they perceived such images had a negative influence, they were asked to consider what help or support they believed adolescent girls might need to support their mental health from social media use, and how they perceived negative influences could be prevented or reduced. For example, questions in interviews included “What are your thoughts about the sexualization of females through images on social media?” (Prompts: is this positive or negative, or both? In what ways?), “What are your perceptions of how such images might make adolescent girls feel and why it might make them feel this way? (Prompts: is this positive or negative, or both)”, “In what ways do you think these feelings might influence adolescent girls’ mental health in positive or negative ways?”, “How do you think adolescent girls can be supported if they are experiencing negative thoughts or feelings?”, and “How do you think any negative thoughts or feelings experienced among girls could be prevented or reduced, or positive thoughts or feelings promoted or enhanced?”.

Two parent interviews were conducted in person (at their home or workplace) and the remaining nine by phone. Four school support service staff and five youth mental health service provider interviews were conducted in person (at their workplace) and three and five, respectively, via phone. All phone interviews were undertaken by participant request. Interviews were conducted for between 30 min and one hour.

### 2.5. Data Analysis

Interviews were audio recorded and professionally transcribed verbatim and imported into qualitative data management software QSR NVivo version 11© to facilitate data analysis, retrieval, and interrogation. Data were deidentified and analyzed thematically, as described by Braun and Clarke [43,44]. This involved the first author reading and re-reading transcripts to generate initial codes emanating from the data, collating the codes into themes using a thematic ‘map’ of the analysis, and interpreting the data set as a whole. Full and equal attention was given to each data item. The coding frame for analysis was guided by inductive codes generated from the study data and deductive codes present in existing research literature [40]. Codes found to not accurately represent the data were amended or erased.

A reflexive approach was adopted by the first author, a cis-gender woman acknowledging that her own biases and background influenced the collection and interpretation of the data. She recognized that although able to relate with participants’ perceptions and experiences, that their views may differ from her own, and that these influence and are influenced by conducting the research.

An audit trail was maintained throughout data collection and analysis by the first author to document comments, decisions, and observations to demonstrate and clarify decision making. This documentation ensured interpretations were accurately reflected in the data and maintained study rigor by strengthening the dependability and confirmability of the research [45,46].

Full ethical approval to conduct this research was obtained from the University of Western Australia Human Research Ethics Committee (RA/4/1/8248).

## 3. Results

The study involved 28 participants (11 parents, 7 school support service staff and 10 youth mental health service providers). Parents were all females aged from 40 to 54 years. Most were university educated (Undergraduate n = 2, post-graduate n = 5) and three had a trade certificate. Most parents had daughters aged 14 or 16 years (n = 4, n = 3). Two parents had daughters aged 15 years, one parented a 17-year-old, and one parented a 14- and 16-year-old daughter. The seven school support service staff participants included five school psychologists, one grade-level coordinator and one head of student support services. Four of these participants were female and three were male, ranging from 26 to over 60 years in age; all were university educated (postgraduate university degree (n = 5); undergraduate degree (n = 2). Two participants worked in government schools and five worked in non-government schools, including four who worked in an all-girls school. Youth mental health service providers were all female and aged between 25 to 64 years. The majority (n = 8) were under 40 years of age. Providers included mental health occupational therapists (n = 8), mental health clinical nurse (n = 1), and youth worker (n = 1). The majority (n = 9) had university qualifications, with three obtaining postgraduate qualifications. Years of youth mental health experience ranged from 3 to over twenty. All participants from each three participant groups spoke English as their first language.

All participants perceived sexualized images typically found on social media as exacerbating poor mental health among adolescent girls. Two interrelated themes emerged related to perceptions of what girls encounter on social media that may influence their mental health; with participants describing the ‘potential for comparison’ and ‘pressure to conform’. Participants also explained how they perceived ‘counteracting negative influences’ could reduce or prevent the potential for mental health harms among girls related to sexualized images on social media and the importance of adults and services ‘keeping up to date’ and being ‘approachable and trustworthy’ when describing the support girls might need.

### 3.1. Potential for Comparison

Participants referred to the potential for adolescent girls to engage in comparisons with sexualized images on social media. They acknowledged that comparisons commonly occur offline but felt they were exacerbated through social media; with girls constantly exposed to sexualized images of females; images depicting girls and women at their ‘best’, not how they look in everyday life. Appearance comparisons were perceived to impact girls’ body image as an indicator of mental health and wellbeing, with Instagram specifically seen to promote and reinforce a focus on appearance;

*I suppose like all parents, I worry about the images that she [daughter] sees [on social media], particularly things like perceptions of body image*.(Parent 9)

*It [Instagram] is just almost completely about looks, you know, it’s pretty rare that you see anything on there [Instagram] that’s about what somebody’s achieved*.(Youth mental health service provider 10)

A few parents discussed the fashion, beauty and make-up related accounts their daughters follow on Instagram. These parents considered such accounts to have a potential positive influence on their daughter, as they did not contain content facilitating comparisons;

*It’s [Instagram account daughter follows] more of tips and hints rather than sexualizing anything. She [the Instagram account] puts a before and after photo up. She will have someone who looks very plain or ordinary and then has her make-up done and just looks absolutely stunning. It kind of shows you that the things you see in the magazines are full-on hair and make-up but if they took all that off, they just look like you and me, ordinary or whatever. Although it’s make-up, it’s not really body image*.(Parent 8)

All participants identified adolescent girls as being more likely to compare themselves to their friends and peers rather than celebrities;

*I think nowadays they’re more influenced by their peers and less by celebrities. At least that’s what I’m hearing from the girls that we see. They’re talking more about comparing themselves to the photos that are posted by peers. I think they’re comparing how many likes they’re getting compared to peers and I don’t know if there’s a certain number that they’re after or there’s something that validates that picture, but I definitely have seen that*.(Youth mental health service provider 5)

*They [girls] compare themselves to each other [on social media]*.(Parent 4)

School support service staff highlighted that they had worked with girls with poor body image and anxiety in their role; specifically, where comparisons made to sexualized photos of females on social media led to the belief their value was attributed to their appearance. In some cases, they referred to how this had contributed to the development of eating disorders among adolescent girls in their schools.

*We’ve had a few eating disorder problems, diagnosis, and that sort of thing. Social media is a contributing factor, how you look is how people rate you as a person and they’re comparing themselves to what girls post of themselves*.(School support service staff 7)

Parents and youth mental health service providers perceived that girls were very self-critical. This, they considered, stemmed from a belief that their value is based on appearance with girls perceiving they were not good enough in comparison to the social media photos of others. One youth mental health service provider describes this:

*I think probably for the bulk of girls, they don’t meet that level of perfection that they feel that everybody else has, you know, has got endless legs and a fantastic figure and they can’t ever meet that. So, I think for a lot of girls, they’re beating themselves up even more, you know, that they’re not pretty enough. It’s so unattainable and so unrealistic by and large*.(Youth mental health service provider 10)

Participants believed some girls are more likely than other girls their age to make comparisons on social media and feel pressure to conform to appearance ideals. School support service staff and youth mental health service providers said body image and self-esteem issues were common among the girls with whom they had worked;

*In my three years on the mental health unit at (public children’s hospital), I would say probably 90% to 95% of the teenage girls that we saw had self-esteem issues in some form and a huge portion of them had body image issues as well. It’s not often their presenting complaint of why they come into hospital but definitely there*.(Youth mental health service provider 7)

*In conversations and talking with them [girls], they might be having some really significant depression or anxiety but underneath that, almost every one of them has some body dissatisfaction or body image issues that are a contributing factor of that more overarching presenting problem*.(School support service staff 7)

Youth mental health service providers specifically believed sexualized images of females typically found on social media exacerbated poor mental health among girls, and while not the cause of mental health difficulties was a contributing factor;

*Have you heard of the stress vulnerability model where you’ve got a bucket and you’re just filling it up with water and it starts overflowing? Every little stressor is a punch in that bucket, you get a hole in that bucket and I think social media and access to all the sexualised images is just another punch in the bucket. I think, already, it’s going to be on top of a lot of other issues*.(Youth mental health service provider 1)

*In terms of anxiety, depression, eating disorders, I think a lot of that stuff can be more recently—I don’t wanna say the social media stuff is the cause of it but it can contribute in a negative way without great strong realisation, I think, from the person that’s what they’re doing. So, not actually aware that their social media usage is actually feeding into their anxiety or their eating disorder or their depression*.(Youth mental health service provider 2)

### 3.2. Pressure to Conform

All participants believed that sexualized images of females on social media created pressure for girls to conform to the females in such images in terms of their appearance and how they present themselves in their own photos on social media;

*[There’s] pressure on teenage girls to be cool. I don’t like it [how girls portray themselves on Instagram], particularly. They do seem to be under pressure to meet a certain look or conform to a certain standard*.(Parent 1)

They also discussed the role of likes and comments on social media; how they contribute to the pressure to conform and impact on mental health;

*They’re seeing what other people are getting likes for, and then they’re getting an idea of what they should put out there and often it’s a certain type of sexualized look that they’re reinforced now of what is beautiful, what is desirable. So, there’s that pressure from there. And then when they maybe don’t get the likes that they would like or the comments that they would like, then that can be really devastating for them*.(School support service staff 7)

*They feel they need to put up a certain front, they need to look a certain way, they need to show themselves a certain way, so that they actually become accepted and become valued through the likes and comments and things like that on social media*.(Youth mental health service provider 9)

All participants perceived girls posting sexualized photos of themselves on social media (commonly described as near-naked photos or posing suggestively and/or pouting) was a result of pressure to conform with peers. Parents spoke about the pressure girls feel to fit in with their friendship groups, and how this played out on social media by posting provocative photos;

*I think the appearance anxiety is more prevalent in the girls than in the boys. I think girls think about these things [their appearance and approval from their friends on social media] a lot more intensely*.(Parent 1)

When discussing the pressure to conform, youth mental health service providers drew from their experience working with girls who they described as having complex and traumatic backgrounds. They perceived this population sub-set was particularly vulnerable to the influence of sexualized images online, further contributing to a desire to be thin and engage in sexualized behaviour on social media, as well as engaging in disordered eating and excessive exercise;

*[I work with] young vulnerable girls who are looking for somewhere to belong. So, they’re more vulnerable, I think, to being sexualized. It’s a group norm for them and you get these girls showing you this very glamorised photo of themselves dressed in a short frock or whatever and they’ve got lots of likes and that’s positive feedback for them. And they already have problems with body image and eating disorders*.(Youth mental health service provider 8)

*There is a bombardment of sexualized images and working with girls with additional trauma and other stuff going on, I can’t detach their history from their sexualized behavior. It definitely leads to eating disorders and wanting to be thin and exercising a lot, that’s often how it starts*.(Youth mental health service provider 1)

Social media was a prevalent contributor to girls’ eating disorders, as described by a youth mental health service provider;

*I can’t think of an eating disorder assessment I’ve done in probably the last five years that hasn’t included a social media aspect. There’s always something. It’s always part of it as a contributing factor*.(Youth mental health service provider 4)

### 3.3. Counteracting Negative Influences

Counteractive strategies previously used or perceived as helpful to reduce the potential for negative influences on girls’ mental health as a result of sexualized images on social media were described by participants.

Youth mental health service providers encouraged girls to follow Instagram accounts that create a positive influence in their feed and enable access to support (i.e., youth mental health focused accounts);

*One of the things we talk about is around following helpful sites on Instagram, so you get them [girls] to follow headspace, What’s Up, those kinds of things, so that they’ve got some positive influences in their feeds and got access to [support] numbers they need*.(Youth mental health service provider 4)

Other strategies they described included developing girls’ resiliency and critical thinking skills in schools and youth mental health settings (particularly for girls experiencing particular vulnerabilities), out of school groups targeting girls to address self-esteem and social media literacy using a combination of education and fun activities, and activities with girls to reflect on their values. One youth mental health service provider spoke about her experience facilitating a value reflection activity with adolescent girls in a group setting in her work;

*[I asked the girls] to think about what they value in a friend, what kind of things they would like in a partner, and after writing them down, asked if anyone could show me where it says beautiful, tall or rich? They had written things like sense of humour, kind, likes animals, and we noted the inconsistency to the things they often aspire to be and in reality, they don’t rate those things that much themselves. They haven’t picked their friends because they have got long blonde hair. They want someone that’s kind and honest and laughs at their jokes*.(Youth mental health service provider 10)

Parents thought negative influences could be counteracted by girls following Instagram accounts they admire for achievements not appearance, such as authors and activists. Other suggestions included education (from primary school age) addressing appreciation for diversity of appearance and gender representations in media, with a focus on body functionality rather than appearance. One parent spoke about her daughter, who was doing competitive ballet, and how she viewed her body and its functionality;

*In terms of her body image, she definitely sees her body as something that she needs to fuel properly in order to do that [ballet] and she needs to care for it in order to be able to do what she wants to do with it*.(Parent 9)

School support service staff discussed body positivity and acceptance messaging in schools (for girls and boys), and targeted education for boys about gender representations and attitudes and behaviour toward girls. They also thought targeted education for girls was needed. Such education could include cognitive behaviour therapy informed exercises in small groups from primary school age—to differentiate thought, behaviour and emotion (i.e., to challenge thoughts of posting photos on social media to look thin or sexy for validation through likes and comments and to encourage alternatives instead such as catching up with friends, going for a walk or watching a movie). A need for social media literacy for students and other school staff was also highlighted, as described by one school psychologist;

*School psychs absolutely have a role in helping staff understanding social media literacy themselves to be able to then pass it on [to students], but it’s whether the school feels it’s a priority or whether the school has that, I guess, philosophical approach to either, “Alright, well how are we going to educate the kids? How to do this?” rather than how we’re going to get rid of social media problems in our school. That’s not going to happen. It’s about helping the students to better prepare for when they are faced with difficulties on social media or with content whether or not they’re sure how they feel about that*.(School support service staff 7)

### 3.4. Keeping up to Date

Participants perceived adults caring for and working with girls need to keep up to date regarding ways to provide support. All participants recognised the constantly and rapidly evolving nature of social media, with challenges in keeping up to date. However, they believed being aware (as best they could) of social media developments, would assist them to support girls to reduce any potential mental health harms;

*Not only do we kind of have to keep up with the times, but it’s important for me to know what she’s [daughter] talking about and what she’s up against because if I don’t know, then I can’t help her*.(Parent 11)

Youth mental health service providers and school support service staff acknowledged they need to keep up to date with changes in social media, as related issues were common among the girls they support. School support service staff reported challenges related to the school setting. They spoke about difficulties in providing the quality of support girls need that depended on the approach of an individual school toward sexualized content on social media;

*I think it’s got to do with how we’re helping these girls, or these young ladies feel good about themselves because there’s clearly an issue which is why they’re going down this route of the sort of social media use because a lot of them aren’t feeling great about stuff at school, not really doing anything about it. They’re [the school] saying don’t do it, don’t share these sort of photos, but I guess if they’re feeling low, or they’re feeling down, or they’re not feeling great about themselves, and for them in some ways that’s a coping mechanism, but it helps them, so maybe that night or the next day and then they stop and go what else can they do to increase their self-esteem, their wellbeing*.(School support service staff 6)

For mental health service providers, social media related items were now included in their mental health assessments;

*I think clinicians are just gonna have to shift and get updated—There’s now a section in our assessment on social media because it’s becoming such an issue, you can’t just add it into other problems. It has its own section now. School has section and when you consider it in a context, home has a section, family has a section, and now social media has its own section. So, our systems just changed*.(Youth mental health service provider 4)

Some youth mental health service providers described instances where girls they were working with would show them photos of themselves posing and pouting in a sexualized way, asking for their advice about posting these on Instagram. They reported feeling unsure of how to respond and did not feel they were up to date on how to discuss social media behaviour with girls;

*One of the girls, she’d pose for a photo, and then instantly be, “What do you think? Look at this. What do you think of this photo? Which one should I keep to put on my Instagram?” Really concerned, “Oh, no one will probably like it anyway.” A real negative sort of, “Oh, you know, I’m not as pretty.” I didn’t know what to say*.(Youth mental health service provider 3)

Most youth mental health service providers also mentioned they felt parents needed regular and current updates about approaches and strategies to foster positive relationships with their daughters; for all parents but particularly parents of girls engaged with mental health services (where relationships were often tense);

*Parents need education around basic connections, particularly mother and daughter, because I think that was a big factor when I was working on the wards. It’s just that parents had no idea what was going on and trying to explain it [the difficulties related to social media their daughters were experiencing] to them, they had no concept of any of it. So, I think that’s a massive thing, so that they can help their daughters to navigate those things [social media] themselves too*.(Youth mental health service provider 7)

*Parents need to step up, but the reality is that all of the girls that we see in mental health services have not been—for lots of reasons, have not been parented. So, there’s a disconnect often between the parent and the child. They [parents] don’t parent because they’re already really traumatized themselves or they’ve got their own substance abuse issues, or they’re completely disconnected, or they expect their daughters to exhibit self-care from a young age. Parents could benefit from support but it’s a complex issue and I don’t know that the parents of our target demographic [girls receiving treatment in youth mental health services] will respond to any sort of global approach*.(Youth mental health service provider 8)

All participating parents indicated they would like to have more current sources of support for their daughters; including where to direct their daughters to seek help (particularly when they did not want to speak to them or if they did not feel equipped to provide support). They also wanted to access support for themselves as a parent, to learn how to support their daughters;

*Not only do we [parents] have to keep up with the times, but it’s important for me to know what they’re [daughters] talking about and what they’re up against [on social media] and where there is support for them, because if I don’t know, then I can’t help them*.(Parent 11)

### 3.5. Approachable and Trustworthy

All participants highlighted the importance of adults being perceived by girls as approachable and trustworthy if they were to seek their help for any mental health difficulties related to social media use. Some parents did not think their daughters would seek support at school because of concerns about confidentiality, while others thought their daughters would seek help from teachers who they found to be approachable and trustworthy;

*I think maybe they’d [girls] be a bit worried about going to the school [for support], really. It could be digging the hole deeper for themselves because then it might need to be brought to the parents’ attention. I don’t know how far it would go with respect to confidentiality*.(Parent 2)

*I think they [schools] should be [a helpful source of support] cos they [girls] spend a lot of hours there. A teacher that they [girls] click with, that you could share things with and trust, I think it could be a first port of call*.(Parent 3)

*She [daughter] does have one teacher at school that she really likes and she does talk to all the time, so maybe a teacher is always good as well. I know there’re some teachers she wouldn’t talk to about anything, but then she’s got one or two teachers that she’s really close with*.(Parent 6)

All parents mentioned an adult was seen as more approachable if they could relate to what girls were experiencing. They suggested schools utilising young females (older but close in age), such as university students, to speak about issues related to mental health and social media use;

*I think it would be great if they [schools] took these kind of young, who can relate, just almost out of university and talking to them [girls], or having guest speakers to come around and just say, “Hey, listen.” I think that would be really helpful at school for young women*.(Parent 6)

Parents also mentioned the importance to girls of being able to remain anonymous when seeking support, so they could disclose any concerns while remaining unknown to the source of support. Online sources were described as providing an ideal setting for girls to remain anonymous while seeking help.

A sentiment reflected in both the school support service staff’s and youth mental health service providers’ interviews, was the importance of a supportive school culture to facilitate help-seeking among students. For one of the participating school psychologists, this was considered crucial in his low socioeconomic school;

*I think working in the context that I do, it’s quite a low socioeconomic area and their [girls] parents may not necessarily always have the most supportive approach to difficulties, and we do have a good team here [at the school] and it’s quite a varied team. A lot of students do come here [student services] as a safe place and safe rapport I guess, which we are lucky but I know that’s not always the case. It depends on the skills of the team, and it depends on the make-up of the team and also school culture. We have a really supportive school culture here in terms of student services support which is good*.(School support service staff 7)

Teamed with a need for anonymity by girls at times, participants also discussed face-to-face contact. They believed it was key to building trust and connection with girls, ensuring they feel comfortable speaking about their mental health from their social media use—as one school support service staff member described;

*The idea of confidentiality probably appeals to a certain extent that they [girls] can talk about it but speaking from our girls’ perspective here [at school], I think we do a pretty good job in having that sense of trust to come to us and we’re not judgemental, that we can help them. There’s something affirming about someone face-to-face, things that you can see, you can read their facial expression, you can talk and you develop some sort of connection, whereas online, I think, it’s a voice or they could be typing or whatever it is, that you haven’t got that personable feel, which I think we all do*.(School support service staff 1)

Parents perceived they would be important sources of support for their daughters, considering their approachability and open communication in their relationships. They acknowledged girls’ need for autonomy and as mothers they wanted to balance being there for their daughter with respecting their independence regarding social media use;


*I think it’s really important just to be able to say to them [daughters], “If you see something that’s really gross or upsetting [on social media], you can tell me, and I’m not gonna freak out. Let’s just have a chat about it.”*
(Parent 11)


*I had a talk with her [daughter] last night and I said, “Do you ever feel sort of down or concerned about any of the images that you see [on social media]? Does it make you feel—in terms of comparison, do you feel lesser?” She said, I’ll look at them and think—oh, she’s got a nice body or I like that dress or I like the way she’s got her makeup done,” but she said, “I don’t really feel bad about it. It’s just how it is.” I said, “That’s great.” And I sort of took the opportunity to say, “Look, if that ever does change and you wanna talk about any of this, you can.”*
(Parent 10)

## 4. Discussion

The current study explored parents of adolescent girls’, school support service staff’s, and youth mental health service providers’ perceptions of how sexualized images typically found on social media might influence adolescent girls’ mental health (in positive or negative ways), what support girls might need should they experience mental health difficulties, and how such difficulties could be prevented or reduced. This study found that all participant groups believed sexualized images on social media can exacerbate poor mental health among adolescent girls. They perceived sexualized images on social media could negatively influence girls’ mental health through an interplay of the potential for comparison and pressure to conform that girls encounter in this environment. The aspects of girls’ mental health that participants believed could be exacerbated by sexualized images on social media were body image and self-esteem, which were also seen to potentially increase the risk of developing eating disorders for some girls. Participants considered it crucial that adults and services keep up to date and be perceived by girls as approachable and trustworthy to enable appropriate for support for those who may experience mental health difficulties. They also discussed the importance they placed on counteracting negative influences of sexualized images on social media as a strategy for preventing or reducing mental health harms among girls.

Consistent with previous research, participants in this study perceived sexualized images could potentially contribute to negative body image among adolescent girls [20]. The potential negative impact on girls’ body image was particularly apparent in discussions with school support service staff and youth mental health service providers, which likely reflects the nature of their role to provide mental health support to girls. Youth mental health service providers explained that social media related items were now included in their mental health assessments, suggesting the extent of influence social media is possibly having in girls’ lives. Based on their experiences working with adolescent girls, youth mental health service providers perceived girls who had complex and traumatic backgrounds were especially vulnerable to the impacts of sexualized images on social media on body image. This perception suggests both the girls from complex and traumatic backgrounds, and the youth mental health service providers who work with them, are potentially important groups to engage with to develop interventions and support strategies related to sexualized images on social media. Additionally, it is important to consider that the participating adults’ perceptions of sexualized images as contributing negatively to adolescent girls’ body image may be reflective of their own experiences and feelings related to sexualization, particularly for female participants.

Sexualized images on social media were seen by all participants as potentially enabling adolescent girls to make negative appearance-based comparisons with possible subsequent negative influences on their body image and self-esteem. In line with previous research conducted with girls [21,47], participants believed comparisons made by girls to the females they see in sexualized images on social media, and the positive feedback they receive through likes and comments, could reinforce value based on appearance. Consequently, participants felt girls might internalize pressure to conform with the females in the sexualized images they view in terms of their appearance and how they present themselves in their own photos on social media. Objectification Theory [23] can provide a useful explanatory framework for the interpretation of these findings. This theory suggests self-objectification is a potential mechanism through which sexualized images of females on social media might influence adolescent girls’ body image [24,25,26]. Self-objectification among girls from viewing sexualized images on social media was evident in participant descriptions in this study. These findings indicate self-objectification is perceived as a prominent issue by adults who care for and work with adolescent girls, who regularly view sexualized images on social media. Although beyond the scope of qualitative research, further research is needed to determine the directionality of self-objectification and girls’ appearance-based comparisons, as it is unclear whether girls who already have body image concerns are more likely to self-objectify and thus compare themselves to sexualized images on social media and conform to the appearance ideals reinforced.

The influential role of peers in reinforcing social norms and normative expectations about how adolescent girls should look and behave on social media in relation to sexualized images was evident in the perceptions of participants in this study. Participants thought girls more often made appearance-based comparisons to sexualized images of their peers on social media than celebrities. This finding aligns with previous research identifying the significant influence of peers on girls’ body image [14,48]. Girls have reported perceiving their peers as more relatable and similar to themselves than celebrities, and as such view peers’ images on social media as more relevant when making appearance-based comparisons [34,48]. Peer appearance-based comparisons may therefore be important to include in the prevention and early intervention of body image concerns among girls related to sexualized images on social media. Furthermore, the findings of this study revealed the perceived role of peers in potentially contributing to adolescent girls experiencing pressure to conform with friends, by posting sexualized images of themselves on social media. The ‘liking’ and ‘commenting’ functions on social media were seen as having the potential to reinforce to girls instantly and visibly what is deemed beautiful and desirable, possibly modelling to them that sharing sexualized images of themselves is an acceptable and valued social norm among their peers that will be rewarded with approval. Social Cognitive Theory [27] provides a useful explanatory framework to interpret these findings. The rewards associated with posting sexualized images may be an important factor determining the extent to which some girls might feel pressured to conform with sharing such images of themselves. This was perceived by participants as a risk factor for girls’ poor body image and low self-esteem, if girls did not receive the anticipated positive feedback on their appearance. When developing interventions and support strategies for adolescent girls, these findings further highlight the need to consider how the girls might be influenced by social norms and normative behaviour among their peer groups and their ability (or inability) to challenge such norms.

The participants in this study considered encouraging adolescent girls to counteract the potential for negative influences from sexualized images on social media as important in preventing or minimizing harms. Parents emphasized a need for girls to follow Instagram accounts of females they admire for achievements, such as authors and activists. Counter to this view though, was a perception among parents that their daughters following fashion, beauty and make-up related Instagram accounts was positive because they featured ‘before and after’ photos of females to show girls the difference make-up and editing can make to appearance. This perception suggests parents may not be able to identify such accounts as appearance focused or their potential for facilitating appearance comparisons among their daughters, with potential negative impacts to body image. Parents, especially mothers, are a major influence on girls’ body image [49,50] and mothers’ self-objectification has been associated with girls as young as 5–8 years of age using beauty products and having appearance concerns [51]. Parents may need body image related education to enable them to help their daughters challenge sexualized appearance-ideals on social media and counteract any potential negative influences, and model behaviours and attitudes that show girls’ value is more than their outward physical appearance [52]. Research has shown parents provide a protective role in preadolescent and adolescent social media appearance comparisons and body dissatisfaction [53,54]. Therefore, when planning health promotion interventions, it is important to consider parents’ level of body image related knowledge and incorporate body image awareness, education and support aimed at adolescent girls’ parents, for possible protective impacts on girls.

Both youth mental health service providers and school support service staff had worked with girls to develop their resiliency and critical thinking skills when using social media. They had done so by using activities girls considered both fun and educational, that also involved girls reflecting on their own values and whether these aligned with the content they view and post on social media. Parents and school support service staff suggested education from primary school age for both females and males about body positivity and acceptance, appreciating diversity of appearance, and with a focus on body functionality rather than appearance, were needed to help counteract potential negative influences from sexualized images on social media among girls. This perception describes components of positive body image, as opposed to negative body image or body dissatisfaction, where one accepts and appreciates the functions of their body regardless of whether it meets dominant societal appearance ideals [55]. Adolescents with positive body image have strongly criticised appearance ideals, a factor found to uphold positive body image through fostering a process known as protective filtering [14,56], where positive-body related information is accepted and negative information rejected [55,57]. However, research has also found adolescent girls’ protective filtering of social media content does not necessarily protect their body image, and that they find it challenging to accept and appreciate their own bodies and internalize body positivity messages [15]. Nevertheless, the perceptions of participants in this study and existing research support the importance of exploring ways to promote positive body image among adolescents when developing interventions aimed at counteracting potential negative influences from sexualized images on social media.

School support service staff highlighted the potential for school psychologists to facilitate social media literacy education in schools, not just for students, but for school staff such as classroom teachers. Social media literacy education is focused on developing skills to examine and critique the underlying messages in images on social media and social interactions among users of social media [37] and has been shown to decrease girls’ body dissatisfaction and appearance comparison and increase self-esteem [58]. Schools are an important setting in which health promotion interventions focused on body image can be implemented [59] and for the provision of training for staff and parents to increase their knowledge, confidence and competence to discuss body image and appearance-ideals on social media and potential impacts with students [60,61]. School psychologists can support the planning, preparation, and delivery of such programs in the school setting and could be an important group to help co-design and engage in school-based prevention and early intervention [62].

The importance of adults being perceived by adolescent girls as approachable and trustworthy was highlighted by participants in this study if girls were to seek their help for any mental health difficulties they were experiencing related to sexualized images on social media. The role of the school as a setting where girls could receive support was discussed by all participants. It was considered crucial for school staff to be perceived as approachable. Schools provide a critical pathway into support for adolescents experiencing mental health difficulties [63]. Parents had mixed views about whether their daughters would seek support at school, but all thought adults in schools were seen as more approachable if they were relatable to students. They suggested schools could work with young females such as university students who were older than students but close in age to speak to girls about issues related to social media and mental health. Empirical evidence supports the facilitation of interventions in schools by undergraduate female university students to increase body appreciation and reduce body dissatisfaction among adolescent girls. This research indicated a high level of acceptability among girls when the Body Project intervention was facilitated by this relatable group [64,65].

School support service staff also perceived a supportive school culture was important in facilitating help-seeking among students and especially for those in low socioeconomic schools. All participants understood girls may sometimes need anonymity when seeking support and believed online sources could be useful to enable this type of support. However, overall, they felt face-to-face contact was vital in building trust and connection with girls to provide them with mental health support. Adolescents may find online sources of support appealing for emotional self-disclosure and social media platforms. This has been found to be useful in providing opportunities for support provision, building trust and improving mental health service access and to link to and complement face-to-face support [66,67].

There are several limitations to the current study that should be considered when interpreting the findings. This study was exploratory and therefore included a small sample of self-selected participants (n = 28). Some participants were recruited through snowball sampling methods. Therefore, findings are not representative of the broader population of parents of adolescent girls or those working in school support service staff or youth mental health service provider roles. However, generalisability is not the purpose of qualitative research which often uses smaller samples to enable the collection of in-depth information and provide direction for further research. Additionally, the triangulation of sampling strategies used in this study may help to reduce the limitations related to snowball sampling methods.

The characteristics of the participants must also be considered. Parents of adolescent girls were all female, aged 40 to 54 years and most were university educated. Additionally, most parents had daughters who were aged 14 or 16 years. Findings may differ among male parents of adolescent girls, younger female parents (who may use social media more than older parents and thus feel more competent speaking with their daughters about social media), those with lower education levels and younger or older daughters. Most school support service staff worked in non-government schools and findings may differ among those who work in government schools. The participating youth mental health service providers were all female, and most were aged 25 to 37 years. As such, findings may vary for males and older females who work in youth mental health. Furthermore, most youth mental health service providers were occupation therapists and there may be differences between this group and other professionals working in youth mental health services. All participants spoke English as their first language; therefore findings may differ among parents, school support service staff and youth mental health service providers from more diverse cultural and linguistic backgrounds.

## 5. Conclusions

The current study contributes new knowledge from the perspective of three adult groups: parents of adolescent girls, school support service staff and youth mental health service providers on how sexualized images of females on social might influence adolescent girls’ mental health, the support girls might need if they experience mental health difficulties, and how such difficulties could be prevented or reduced.

The findings of this study suggest that the participating parents, school support service staff and youth mental health service providers perceive sexualized images on social media as exacerbating poor mental health among girls, contributing to negative body image and low self-esteem through the potential for comparison and pressure to conform experienced by girls. The participants in this study suggested self-objectification and social norms might be important considerations when exploring the role of sexualized images on social media on girls’ mental health. Participants identified strategies they believed could counteract what they perceived as the negative influence of sexualized images on social media. These have important implications for the development of health promotion programs addressing social media use and body image and self-esteem among girls for prevention and early intervention to minimize potential harms. The adults in girls’ lives need to keep up-to-date and be considered approachable and trustworthy if girls are to seek their support for their mental health. Findings highlight the importance of gaining parent, school support service staff and youth mental health service providers’ views to enable an ecological intervention approach. Further research should engage with both adolescent girls and the adults in their lives to identify and investigate the effectiveness of strategies to prevent and reduce girls’ experiences of negative body image and low self-esteem associated with sexualized images on social media.

## Data Availability

Data are available from the first author upon reasonable request.

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
