# Peer review of "Sexualized Images on Social Media and Adolescent Girls’ Mental Health: Qualitative Insights from Parents, School Support Service Staff and Youth Mental Health Service Providers"

_ijerph, 2022, doi:10.3390/ijerph20010433_

Round 1

Reviewer 1 Report

This is very compelling and well written study. I would only suggest a few minor amendments:

Introduction: Even though all the research covered in this section is relevant to the chosen topic, I believe readers would benefit from reading a little bit more about previous qualitative research around the area (even if, as stated by the authors, there is no previous research addressing the specific research question posed). The Introduction reads a little bit too quantitative for a qualitative research project. 

Method:

A few examples of the exploratory questions used for the interviews would have been useful to replicate the results.

Have the authors considered incorporating the latest amendments to the TA approach proposed by Braun and Clarke? Only her 2006 paper seems to be cited in this manuscript. See for example https://www.tandfonline.com/doi/abs/10.1080/2159676X.2019.1628806?journalCode=rqrs21 

A reflexivity subsection (either here or under the Discussion section) would be really useful.

General comment: Please avoid sexist language in your work. Replace terms like "his/her" by 'them'. The APA guidelines for anti-sexist language could be really useful for a full review of the article.

Reviewer 2 Report

There is insufficient distinction between the adults' (parents, school support workers, and mental healthcare workers) anxieties and concerns about what might be harmful to teen girls, and what is demonstrably harmful to girls. The (mostly female) adults' anxieties may reflect their own vulnerability to gendered oppression and feelings of relative helplessness to effectively protect the youth they care about from the forces they (adults) fear.

The article reproduces relatively long standing dominant medical/social science discourse (theory rather than evidence) about the causes of dangerously low self-esteem among girls, e.g. vulnerability to sexualized media and objectification, and poor mothering. These may be causal factors, but the study reports the belief (amongst the interview subjects) that these are important causal factors rather than demonstrating that they are, and then offers solutions based on the assumption that we know these are the most important causal factors.

Part of the discourse is that girls and women are manipulated by media, rather than accurately learning about real social forces through these media. Telling them not to pay attention to media that emphasizes women's appearance may be telling them to deny their accurate perception of the facts of what is socially valued (no matter how much we believe it should not be the basis of girls and women's worth). This may undermine the trust that the authors and their adult subjects identified as vital to supporting girls' mental health.

A report on the attitudes and beliefs of the adults' interviewed is interesting. It should not be conflated with an empirically demonstrated analysis of what is threatening to girls' mental health and how to best support it.

It would be useful to indicate other characteristics of the subject pool that may be relevant to anxieties about sexualized social media. Do the subjects identify as cis-gendered? Are school and mental health workers also parents, of girls? Are the parents, parents of teens who identify as cis-gendered? Do the adults or the children they parent or work with identify as racialized, dis/abled, or otherwise marked in ways that may make them especially vulnerable to pressures to conform to dominant norms of attractiveness and social worth, or give them the advantage of critical perspective on them?

Reviewer 3 Report

Title: Sexualized images on social media and adolescent girls’ mental health: Qualitative insights from parents, school support service staff and youth mental health service providers

1.       The study focuses on the influence of sexualized images on social media on adolescent girls’ mental health as perceived by the parents, school support service staff and mental health service providers.

2.       The research design is appropriate for the current study as it examines in broad strokes the social circumstances affecting adolescent girls’ mental health regarding sexual images on social media.

3.       The discussion is very robust and is supported by the Results presented in the study.

4.       I just wonder why only female parents were chosen when it is important also to know how male parents perceive the influence of sexualized images on social media.  While a feminine insight is in order, a masculine insight might add a different perspective.

5.       Verb tense should be consistent.

6.       Literature review may be updated given the recent studies dwelling on related topic.
